# Improved techniques for deterministic $l_2$ robustness

**Sahil Singla**

Department of Computer Science
University of Maryland
ssingla@umd.edu

**Soheil Feizi**

Department of Computer Science
University of Maryland
sfeizi@umd.edu

## Abstract

Training convolutional neural networks (CNNs) with a strict 1-Lipschitz constraint under the $l_2$ norm is useful for adversarial robustness, interpretable gradients and stable training. 1-Lipschitz CNNs are usually designed by enforcing each layer to have an orthogonal Jacobian matrix (for all inputs) to prevent the gradients from vanishing during backpropagation. However, their performance often significantly lags behind that of heuristic methods to enforce Lipschitz constraints where the resulting CNN is not *provably* 1-Lipschitz. In this work, we reduce this gap by introducing (a) a procedure to certify robustness of 1-Lipschitz CNNs by replacing the last linear layer with a 1-hidden layer MLP that significantly improves their performance for both standard and provably robust accuracy, (b) a method to significantly reduce the training time per epoch for Skew Orthogonal Convolution (SOC) layers ($> 30\%$ reduction for deeper networks) and (c) a class of pooling layers using the mathematical property that the $l_2$ distance of an input to a manifold is 1-Lipschitz. Using these methods, we significantly advance the state-of-the-art for standard and provable robust accuracies on CIFAR-10 (gains of $+1.79\%$ and $+3.82\%$) and similarly on CIFAR-100 ($+3.78\%$ and $+4.75\%$) across all networks. Code is available at https://github.com/singlasahil14/improved_l2_robustness.

## 1 Introduction

The Lipschitz constant [1] of a neural network $f : \mathbb{R}^d \to \mathbb{R}^c$, denoted by $\text{Lip}(f)$, controls the change in the output divided by change in the input (both changes measured in the $l_2$ norm). Previous work provides evidence that a small Lipschitz constant is useful for interpretable saliency maps [Tsipras et al., 2018], generalization bounds [Long and Sedghi, 2020], Wasserstein distance estimation [Villani, 2008], adversarial robustness [Szegedy et al., 2014] and preventing gradient explosion during backpropagation [Xiao et al., 2018]. Several prior works [Miyato et al., 2018, Gulrajani et al., 2017] use heuristic methods to enforce Lipschitz constraints to successfully address problems such as stabilizing GAN training. However, these methods do not enforce a guaranteed lipschitz bound and it remains challenging to achieve strong results with provable guarantees.

Using the composition property i.e. $\text{Lip}(g \circ h) \leq \text{Lip}(g)\,\text{Lip}(h)$, we can construct a 1-Lipschitz neural network by constraining each layer to be 1-Lipschitz. However, a key difficulty with this approach is that because a 1-Lipschitz layer can only reduce the gradient norm during backpropagation, for deeper networks, this results in small gradient values for layers closer to the input making training slow and difficult. To address this problem, Anil et al. [2018] introduce Gradient Norm Preserving (GNP) architectures where each layer preserves the gradient norm during backpropagation. For 1-Lipschitz Convolutional Neural Networks (CNNs), this involves using orthogonal convolutions (convolution layers with an orthogonal Jacobian matrix) [Li et al., 2019b, Trockman and Kolter,

---

[1]In this work, we assume the Lipschitz constant under the $l_2$ norm.

Table 1: CIFAR-10 results using faster gradients, projection pooling, CRC-Lip

| SOC layers | Time/epoch (secs) | | Reduction | Standard accuracy | | Provable robust accuracy | |
|---|---|---|---|---|---|---|---|
| | Ours | Previous | | Ours | Previous | Ours | Previous |
| 6 | **25.34** | 30.63 | **-17.27%** | **79.36%** | 76.68% | **67.13%** | 60.09% |
| 11 | **37.11** | 48.44 | **-23.39%** | **79.57%** | 77.73% | **66.75%** | 62.82% |
| 16 | **49.27** | 66.74 | **-26.17%** | **79.44%** | 77.78% | **66.99%** | 62.75% |
| 21 | **61.59** | 83.18 | **-25.96%** | **79.13%** | 77.50% | **66.45%** | 63.31% |
| 26 | **71.51** | 100.70 | **-28.99%** | **79.19%** | 77.18% | **66.28%** | 62.46% |
| 31 | **84.00** | 119.55 | **-29.74%** | **78.64%** | 74.43% | **66.05%** | 59.65% |
| 36 | **95.06** | 137.10 | **-30.66%** | **78.57%** | 72.73% | **65.94%** | 57.18% |
| 41 | **106.01** | 156.25 | **-32.16%** | **78.41%** | 71.33% | **65.51%** | 55.74% |

2021, Singla and Feizi, 2021, Yu et al., 2022, Kiani et al., 2022] and using a class of GNP activation functions called HouseHolder activations [Singla et al., 2022].

Among orthogonal convolutions, SOC [Singla and Feizi, 2021] achieves state-of-the-art results on CIFAR-10, 100 [Singla et al., 2022]. SOC uses the following two mathematical properties to construct an orthogonal convolution: (a) the exponential of a skew symmetric matrix is orthogonal and (b) the matrix exponential can be computed using the exponential series. A drawback of using the exponential series is that it requires us to apply the convolution operation multiple times to achieve a reasonable approximation of an orthogonal matrix. Consequently, during backpropagation, computing the gradient with respect to the weights for a single SOC layer requires us to compute the weight gradient for each of these convolutions resulting in significant training overhead. In this work, we show that because the matrix is skew symmetric, using some approximations, the gradients with respect to weights can be computed in significantly reduced time with no loss in performance. This enables us to reduce the training time per epoch using SOC by $> 30\%$ for networks with large number of SOC layers (Results in Table 1).

Another limitation of 1-Lipschitz CNNs is that their performance is often significantly below compared to that of standard CNNs. Recently, Singla et al. [2022] introduced a procedure for certifying robustness by relaxing the orthogonality requirements of the last linear layer (i.e. the linear layer mapping penultimate neurons to class logits) achieving state of the art results on CIFAR-10, 100 [Krizhevsky, 2009]. Since MLPs are more expressive than linear layers, one would expect improved *standard accuracy* by replacing the last linear layer with them. However, the resulting networks are not 1-Lipschitz and achieving high robustness guarantees (*provable robust accuracy*) is difficult.

To certify robustness for these networks, note that since the mapping from input to penultimate layer is 1-Lipschitz, the robustness certificate for the penultimate output also provides a certificate for the input. Thus, we first replace the linear layer mapping penultimate output to logits with a 1-hidden layer MLP (Multi layer Perceptron) because such MLPs are easier to certify compared to deep MLPs. To certify robustness for the MLP, we use the Curvature-based Robustness Certificate or CRC [Singla and Feizi, 2020]. To train MLP to have high robustness guarantees, we use a variant of adversarial training that only applies adversarial perturbations to the MLP (not the whole network). We call our certification procedure CRC-Lip. This results in improved results for both the standard and also the provable robust accuracy across all network architectures on CIFAR-10 ($\geq 1.63\%$, $\geq 3.14\%$) and CIFAR-100 ($\geq 2.49\%$, $\geq 2.27\%$ respectively). Results are in Tables 3 and 4.

While several works have attempted to construct novel and more expressive orthogonal convolution layers and GNP activation functions, current state-of-the-art 1-Lipschitz CNNs still use pooling layers by taking the max of different elements. In this work, we introduce a class of 1-Lipschitz pooling layers called *projection pooling* using the following mathematical property: Given a manifold $\mathcal{M} \subset \mathbb{R}^n$ and input $\mathbf{x} \in \mathbb{R}^n$, the function $d_{\mathcal{M}} : \mathbb{R}^n \to \mathbb{R}$ defined as the $l_2$ distance of $\mathbf{x}$ to $\mathcal{M}$ is 1-Lipschitz. Thus, to construct a pooling layer, we can first define a *learnable manifold* with parameters $\Theta$ (Example in Section 6). During training, for input $\mathbf{x} \in \mathbb{R}^n$, the pooling layer simply outputs $d_{\mathcal{M}}(\mathbf{x}) \in \mathbb{R}$ as the output, resulting in decrease in input dimension by a factor of $n$. Moreover, since the output $d_{\mathcal{M}}(\mathbf{x})$ is also a function of $\Theta$, $\Theta$ can be learned during training. However, solving for the distance $d_{\mathcal{M}}(\mathbf{x})$ can be difficult especially when $\mathcal{M}$ is a high-dimensional manifold.

To address this limitation, (a) we use 2D projection pooling layers that reduce the dimension by factor of 2 and (b) we construct these layers using piecewise linear curves for which the distance can be computed efficiently by computing the minimum distance to all line segments and the connecting points (Example in Appendix Figure 2). If the curve is closed and without self-intersections, we can also define a signed projection pooling for which the signs of the output for points inside and outside the region enclosed by the curve are different ($\mathbf{x}$ and $\mathbf{y}$ in Figure 2). This allows the subsequent layers to distinguish between the inputs inside and outside the region. In this work, we show some preliminary results using a simple 2D projection pooling layer (Section 6). We leave the problem of constructing high performance 2D projection pooling layers open for future research.

In summary, in this paper, we make the following contributions:

- We introduce a method for faster computation of the weight gradient for SOC layers. For deeper networks, we observe reduction in training time per epoch by $> 30\%$ (Table 1).

- We introduce a certification procedure called CRC-Lip that replaces the last linear layer with a 1-hidden layer MLP and results in significantly improved standard and provable robust accuracy. For deeper networks ($\geq 35$ layers), we observe improvements of $\geq 5.84\%, \geq 8.00\%$ in standard and $\geq 8.76\%, \geq 8.65\%$ in provable robust accuracy ($l_2$ radius $36/255$) on CIFAR-10,100 respectively.

- We introduce a large class of 1-Lipschitz pooling layers called *projection pooling* using the mathematical property that the $l_2$ distance $d_{\mathcal{M}}(\mathbf{x})$ of an input $\mathbf{x}$ to the manifold $\mathcal{M}$ is 1-Lipschitz.

- On CIFAR-10, across all architectures, we achieve the best standard and provable robust accuracy (at $36/255$) of $79.57, 67.13\%$ respectively (gain of $+1.79\%, +3.82\%$ from prior works). Similarly, on CIFAR-100, we achieve $51.84, 39.27\%$ ($+3.78\%, +4.75\%$ from prior works). These results establish new state-of-the-art results in the standard and provable robust accuracy on both datasets.

## 2  Related work

**Provable defenses against adversarial examples**: For a provably robust classifier, we can guarantee that its predictions remain constant within some region around the input. Most of the existing methods for provable robustness either bound the Lipschitz constant or use convex relaxations [Singh et al., 2017, 2018a, 2019a,b,c, Weng et al., 2018, Salman et al., 2019b, Zhang et al., 2019, 2018, Wong et al., 2018, Wong and Kolter, 2018, Singh et al., 2018b, Raghunathan et al., 2018, Dvijotham* et al., 2018, Croce et al., 2019, Gowal et al., 2019, Dvijotham* et al., 2020, Lu and Kumar, 2020, Singla and Feizi, 2020, Bunel et al., 2020, Leino et al., 2021, Leino and Fredrikson, 2021, Zhang et al., 2021, 2022, Wang et al., 2021, Huang et al., 2021, Müller et al., 2021, Singh et al., 2021, Palma et al., 2021]. However, these methods are often not scalable to large neural networks while achieving high performance. In contrast, randomized smoothing [Liu et al., 2018, Cao and Gong, 2017, Lécuyer et al., 2018, Li et al., 2019a, Cohen et al., 2019, Salman et al., 2019a, Levine et al., 2019, Kumar et al., 2020a,b, Salman et al., 2020, 2021] scales to large neural networks but is a *probabilistically certified defense*: certifying robustness with high probability requires generating a large number of noisy samples leading to high inference-time. The defense we propose in this work is deterministic and not comparable to randomized smoothing.

**Provably Lipschitz neural networks**: The class of Gradient Norm Preserving (GNP) and 1-Lipschitz fully connected neural networks was first introduced by Anil et al. [2018]. To design each layer to be GNP, they orthogonalize weight matrices and use a class of piecewise linear GNP activations called GroupSort. Later, Singla et al. [2022] proved that for any piecewise linear GNP function to be continuous, different Jacobian matrices in a neighborhood must change via householder transformations, implying that GroupSort is a special case of more general HouseHolder activations. Several previous works enforce Lipschitz constraints on convolution layers using spectral normalization, clipping or regularization [Cissé et al., 2017, Tsuzuku et al., 2018, Qian and Wegman, 2019, Gouk et al., 2020, Sedghi et al., 2019]. However, these methods either enforce loose lipschitz bounds or do not scale to large networks. To ensure that the Lipschitz constraint on convolutional layers is tight, recent works construct convolution layers with an orthogonal Jacobian [Li et al., 2019b, Trockman and Kolter, 2021, Singla and Feizi, 2021, Yu et al., 2022, Su et al., 2022, Kiani et al., 2022]. These approaches avoid the aforementioned issues and allow training of large, provably 1-Lipschitz CNNs achieving state-of-the-art results for deterministic $l_2$ robustness.

# 3 Problem setup and Notation

For a vector $\mathbf{v}$, $\mathbf{v}_j$ denotes its $j^{th}$ element. For a matrix $\mathbf{A}$, $\mathbf{A}_{j,:}$ and $\mathbf{A}_{:,k}$ denote the $j^{th}$ row and $k^{th}$ column respectively. Both $\mathbf{A}_{j,:}$ and $\mathbf{A}_{:,k}$ are assumed to be column vectors (thus $\mathbf{A}_{j,:}$ is the transpose of $j^{th}$ row of $\mathbf{A}$). $\mathbf{A}_{j,k}$ denotes the element in $j^{th}$ row and $k^{th}$ column of $\mathbf{A}$. $\mathbf{A}_{:j,:k}$ denotes the matrix containing the first $j$ rows and $k$ columns of $\mathbf{A}$. We define $\mathbf{A}_{:j} = \mathbf{A}_{:j,:}$ and $\mathbf{A}_{j:} = \mathbf{A}_{j:,:}$. Similar notation applies to higher order tensors. $\mathbf{I}$ denotes the identity matrix, $\mathbb{R}$ to denote the field of real numbers. We construct a 1-Lipschitz neural network, $f : \mathbb{R}^d \to \mathbb{R}^c$ ($d$ is the input dimension, $c$ is the number of classes) by composing 1-Lipschitz convolution layers and GNP activation functions. We often use the abbreviation $f_i - f_j : \mathbb{R}^d \to \mathbb{R}$ to denote the function so that:

$$(f_i - f_j)(\mathbf{x}) = f_i(\mathbf{x}) - f_j(\mathbf{x}), \qquad \forall\, \mathbf{x} \in \mathbb{R}^d$$

For a matrix $\mathbf{A} \in \mathbb{R}^{q \times r}$ and a tensor $\mathbf{B} \in \mathbb{R}^{p \times q \times r}$, $\overrightarrow{\mathbf{A}}$ denotes the vector constructed by stacking the rows of $\mathbf{A}$ and $\overrightarrow{\mathbf{B}}$ by stacking the vectors $\overrightarrow{\mathbf{B}_{j,:,:}}$, $j \in [p-1]$ so that:

$$\left(\overrightarrow{\mathbf{A}}\right)^T = \left[\mathbf{A}_{0,:}^T,\ \mathbf{A}_{1,:}^T,\ \dots,\ \mathbf{A}_{q-1,:}^T\right], \quad \left(\overrightarrow{\mathbf{B}}\right)^T = \left[\left(\overrightarrow{\mathbf{B}_{0,:,:}}\right)^T,\ \left(\overrightarrow{\mathbf{B}_{1,:,:}}\right)^T,\ \dots,\ \left(\overrightarrow{\mathbf{B}_{p-1,:,:}}\right)^T\right]$$

For a 2D convolution filter, $\mathbf{L} \in \mathbb{R}^{p \times q \times r \times s}$ and input $\mathbf{X} \in \mathbb{R}^{q \times n \times n}$, we use $\mathbf{L} \star \mathbf{X} \in \mathbb{R}^{p \times n \times n}$ to denote the convolution of filter $\mathbf{L}$ with $\mathbf{X}$. We use the the notation $\mathbf{L} \star^i \mathbf{X} \triangleq \mathbf{L} \star^{i-1} (\mathbf{L} \star \mathbf{X})$ and $\mathbf{L} \star^0 \mathbf{X} = \mathbf{X}$. Unless specified, we assume zero padding and stride 1 in each direction.

# 4 Faster gradient computation for Skew Orthogonal Convolutions

Among the existing orthogonal convolution layers in the literature, Skew Orthogonal Convolutions (or SOC) by Singla and Feizi [2021] achieves state-of-the-art results on CIFAR-10,100 [Singla et al., 2022]. SOC first constructs a convolution filter $\mathbf{L}$ whose Jacobian is skew-symmetric. This is followed by a convolution exponential [Hoogeboom et al., 2020] operation. Since the exponential of a skew-symmetric matrix is orthogonal, the Jacobian of the resulting layer is an orthogonal matrix.

However, a drawback of this procedure is that convolution exponential requires multiple convolution operations per SOC layer to achieve a reasonable approximation of the orthogonal Jacobian. Consequently, if we use $k$ convolution operations in the SOC layer during forward pass, we need to compute the gradient with respect to the weights $\mathbf{L}$ (called *convolution weight gradient*) per convolution operation ($k$ times) which can lead to slower training time especially when the number of SOC layers is large. To address this limitation, we show that even if we use $k$ convolutions in the forward pass of an SOC layer, the weight gradient for the layer can be computed using a *single convolution weight gradient* during backpropagation leading to significant reduction in training time.

For simplicity, let us first consider the case of an orthogonal fully connected layer (i.e. not convolution) with the same input and output size ($n$). Later, we will see that our analysis leads to improvements for orthogonal convolutional layers. Further, assume that the weights i.e. $\mathbf{A} \in \mathbb{R}^{n \times n}$ are skew-symmetric i.e. $\mathbf{A} = -\mathbf{A}^T$ and given the input $\mathbf{x} \in \mathbb{R}^n$, the output $\mathbf{z} \in \mathbb{R}^n$ is computed as follows:

$$\mathbf{z} = \left(\sum_{i=0}^{k} \frac{\mathbf{A}^i}{i!}\right)\mathbf{x} + \mathbf{b}, \qquad \text{where} \quad \mathbf{A} = -\mathbf{A}^T \tag{1}$$

We approximate the exponential series: $\exp(\mathbf{A}) = \sum_{i=0}^{\infty} \mathbf{A}^i / i!$ using a finite number of terms ($k$).
**Forward pass** ($\mathbf{z}$): To compute $\mathbf{z}$ during the forward pass, we use the following iterations:

$$\mathbf{u}^{(i)} = \begin{cases} \mathbf{x} & i = k-1 \\ \mathbf{x} + (\mathbf{A}\mathbf{u}^{(i+1)})/(i+1) & i \le k-2 \end{cases} \tag{2}$$

It can be shown that $\mathbf{z} = \mathbf{u}^{(0)}$ (Appendix C). During backpropagation, given the gradient of loss $\ell$ w.r.t. layer output $\mathbf{z}$ i.e. $\nabla_{\mathbf{z}} \ell$, we want to compute $\nabla_{\mathbf{x}} \ell$ (input gradient) and $\nabla_{\mathbf{A}} \ell$ (weight gradient).

**Input gradient** ($\nabla_{\mathbf{x}} \ell$): To compute $\nabla_{\mathbf{x}} \ell$, observe that $\mathbf{z}$ is a linear function of $\mathbf{x}$. Thus, using the chain rule, skew-symmetricity ($\mathbf{A}^T = -\mathbf{A}$) and the property $\left(\mathbf{A}^i\right)^T = \left(\mathbf{A}^T\right)^i$, we have:

$$\nabla_{\mathbf{x}} \ell = \left(\sum_{i=0}^{k} \frac{\mathbf{A}^i}{i!}\right)^T (\nabla_{\mathbf{z}} \ell) = \left(\sum_{i=0}^{\infty} \frac{(-\mathbf{A})^i}{i!}\right)(\nabla_{\mathbf{z}} \ell)$$

We can again approximate the exponential series using the same number of terms as in the forward pass i.e. $k$. To compute the finite term approximation, we use the following iterations:

$$\mathbf{v}^{(i)} = \begin{cases} \nabla_{\mathbf{z}} \ell & i = k-1 \\ \nabla_{\mathbf{z}} \ell - (\mathbf{A}\mathbf{v}^{(i+1)})/(i+1) & i \leq k-2 \end{cases} \tag{3}$$

Similar to forward pass, it can be shown that $\nabla_{\mathbf{x}} \ell = \mathbf{v}^{(0)}$ (Appendix C).

**Weight gradient** ($\nabla_{\mathbf{A}} \ell$): We first derive the exact expression for $\nabla_{\mathbf{A}} \ell$ in the Theorem below:

**Theorem 1** *The gradient of the loss function $\ell$ w.r.t $\mathbf{A}$ i.e. $\nabla_{\mathbf{A}} \ell$ is given by:*

$$\nabla_{\mathbf{A}} \ell = -\sum_{i=1}^{k} \left( \left(\mathbf{A}^{i-1}\mathbf{x}\right) \left(\mathbf{v}^{(i)}\right)^{T} - \mathbf{v}^{(i)} \left(\mathbf{A}^{i-1}\mathbf{x}\right)^{T} \right) \tag{4}$$

*where $\mathbf{v}^{(i)}$ is defined as in equation (3).*

Note that the first outer product i.e. $\left(\mathbf{A}^{i-1}\mathbf{x}\right) \left(\mathbf{v}^{(i)}\right)^{T}$ and the second i.e. $\left(\mathbf{v}^{(i)}\right) \left(\mathbf{A}^{i-1}\mathbf{x}\right)^{T}$ are transpose of each other implying that each term in the summation is skew-symmetric. Although these outer products can be computed in a straightforward way for orthogonal *fully connected* layers, this is not the case for orthogonal *convolution* layers (SOC in this case). This is because, for a convolution filter $\mathbf{L} \in \mathbb{R}^{p \times q \times r \times s}$, the term analogous to $\mathbf{A}^{i-1}\mathbf{x}$ is a patch of size $q \times r \times s$ and that analogous to $\mathbf{v}^{(i)}$ is another patch of size $p \times 1 \times 1$ resulting in an outer product of the desired size $p \times q \times r \times s$ *per patch*. Thus, for SOC layers, each term inside the summation is computed by summing over the outer products for all such patches. For large input sizes, the number of such patches can often be large, making this computation expensive. To address this limitation, we use the following approximation:

$$\nabla_{\mathbf{A}} \ell \approx - \left( \mathbf{u}^{(1)} \left(\mathbf{v}^{(1)}\right)^{T} - \mathbf{v}^{(1)} \left(\mathbf{u}^{(1)}\right)^{T} \right) \tag{5}$$

In Appendix B, we show that the above approximation is principled because after subtracting the exact and approximation gradients (equations (4) and (5)) and simplifying, each term in the resulting series is divided by a large value in its denominator ($\approx 0$). This approximation is useful because in equation (5), the outer product needs to be computed once whereas in equation (4), the outer products need to be computed $k$ times. Also, both $\mathbf{u}^{(1)}$ and $\mathbf{v}^{(1)}$ are computed while computing $\mathbf{u}^{(0)} = \mathbf{z}$ during the forward pass and $\mathbf{v}^{(0)} = \nabla_{\mathbf{x}} \mathbf{z}$ during the backward pass using the recurrences in equations (2) and (3). Thus, we can simply store $\mathbf{u}^{(1)}, \mathbf{v}^{(1)}$ during the forward, backward pass respectively so that $\nabla_{\mathbf{A}} \ell$ can be computed directly using equation (5). In our experiments, we observe that this leads to significant reduction in training time with almost no drop in performance (Tables 1, 3, 4).

## 5 Curvature-based Robustness Certificate

A key property of 1-Lipschitz CNNs is that the output of each layer is 1-Lipschitz with respect to the input. Given an input $\mathbf{x} \in \mathbb{R}^d$, consider the penultimate output $g(\mathbf{x}) \in \mathbb{R}^m$ and logits $f(\mathbf{x}) \in \mathbb{R}^c$ for some 1-Lipschitz CNN. Existing robustness certificates [Li et al., 2019b, Singla et al., 2022] rely on the *linearity* of the function from $g(\mathbf{x}) \rightarrow f(\mathbf{x})$. However, since MLPs have higher expressive power than linear functions [Cybenko, 1989], one way to improve performance could be to replace this mapping with MLPs. However, certifying robustness for deep MLPs is difficult.

To address these challenges, we first replace the mapping from $g(\mathbf{x}) \rightarrow f(\mathbf{x})$ with a 1-hidden layer MLP because they are easier to certify compared to deeper networks. Because computing exact certificates for ReLU networks is known to be NP-complete [Sinha et al., 2018], we use the differentiable $\mathrm{Softplus}$ activation [Dugas et al., 2000] to certify robustness. To certify robustness, we use the Curvature-based Robustness Certificate or CRC [Singla and Feizi, 2020] because it provides exact certificates for a significant fraction of inputs for shallow MLPs. We provide a brief review of CRC in Appendix Section D. Let $g : \mathbb{R}^d \rightarrow \mathbb{R}^m$ be a 1-Lipschitz continuous function and $h : \mathbb{R}^m \rightarrow \mathbb{R}^c$ be a 1-hidden layer MLP such that $f = h \circ g$. Further, let $l$ be the predicted class for input $\mathbf{x}$ i.e. $f_l(\mathbf{x}) \geq \max_{i \neq l} f_i(\mathbf{x})$. Since $g$ is 1-Lipschitz, it can be shown that if $h$ is provably robust in an $l_2$ radius $R$ around input $g(\mathbf{x})$, then $f$ is also provably robust in the $l_2$ radius $R$ around $\mathbf{x}$. The resulting procedure is called CRC-Lip and is given in the following proposition:

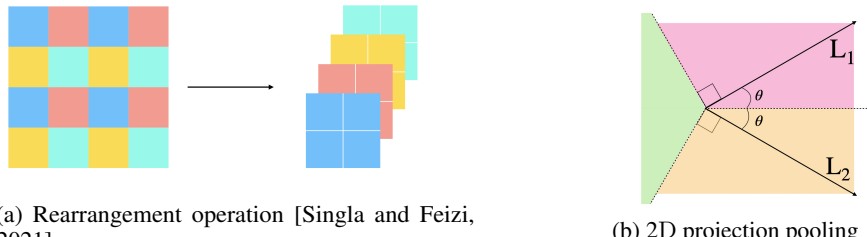

(a) Rearrangement operation [Singla and Feizi, 2021]

(b) 2D projection pooling

Figure 1: Illustration of the rearrangement operation (left) and 2D projection pooling (right).

**Proposition 1** (CRC-Lip) *For input* $\mathbf{x}$ *such that* $f_l(\mathbf{x}) \geq \max_{i \neq l} f_i(\mathbf{x})$, *let* $R$ *be the robustness certificate for* $h$ *using input* $g(\mathbf{x})$, *then* $R$ *is also the robustness certificate for the function* $f$:

$$\min_{i \neq l} \ \min_{h_l(\mathbf{y}^*)=h_i(\mathbf{y}^*)} \|\mathbf{y}^* - g(\mathbf{x})\|_2 \geq R \implies \min_{i \neq l} \ \min_{f_l(\mathbf{x}^*)=f_i(\mathbf{x}^*)} \|\mathbf{x}^* - \mathbf{x}\|_2 \geq R \qquad (6)$$

Proof is in Appendix A.2. In our experiments, we find that although replacing the linear layer with an MLP achieves high standard accuracy, directly using the above certificate often leads to very small robustness certificates and thus low certified robust accuracy. To address this problem, we introduce (a) an adversarial training procedure that *only* applies to the input of the MLP $g(\mathbf{x})$ (i.e. not the input of the neural network $\mathbf{x}$) and (b) curvature regularization to reduce the curvature of the MLP. It was shown in Singla and Feizi [2021] that combining adversarial training with curvature regularization leads to significantly improved provable robust accuracy with small reduction in standard accuracy. This results in the following loss function for training:

$$\min_{\Omega, \ \Psi} \quad \mathbb{E}_{(\mathbf{x},l)\sim\mathcal{D}} \left[ \left( \max_{\|\mathbf{y}^*-g_\Psi(\mathbf{x})\|_2 \leq \rho} \ell\left(h_\Omega(\mathbf{y}^*), l\right) \right) + \gamma \, \mathcal{K}_h \right] \qquad (7)$$

In the above equation, $\Omega$ denote the parameters of the MLP (i.e. $h$), $\Psi$ are the parameters of the 1-Lipschitz function $g$, $\mathcal{K}_h$ is the bound on the curvature of the MLP, $\rho$ denotes the $l_2$ perturbation radius and $\gamma$ denotes the curvature regularization coefficient. The curvature bound $\mathcal{K}_h$ is the same as in Singla and Feizi [2020]. Also, since we apply adversarial perturbations in the penultimate layer, we also need to backpropagate through this procedure to enable training of previous layers. To this end, we simply use an identity map (same gradient output as the gradient input) to backpropagate through the adversarial training procedure and find that it works well in practice, achieving significantly better results compared to the state-of-the-art. It is possible that a more principled method of backpropagation may lead to better results and we leave that avenue open for future research.

## 6  Projection pooling layers

In 1-Lipschitz CNNs, pooling is usually carried out as follows: given input $\mathbf{X} \in \mathbb{R}^{q \times r \times r}$ ($r$ is even), we first use *rearrangement* [Jacobsen et al., 2018] illustrated in Figure 1b to construct $\mathbf{X}' \in \mathbb{R}^{4q \times (r/2) \times (r/2)}$. Next, we apply an orthogonal convolution which gives an output of the same size $\mathbf{Z} \in \mathbb{R}^{4q \times (r/2) \times (r/2)}$ and divide it into two tensors of equal sizes (along the channel dimension) giving $\mathbf{Z}_{:2q} \in \mathbb{R}^{2q \times (r/2) \times (r/2)}$ and $\mathbf{Z}_{2q:} \in \mathbb{R}^{2q \times (r/2) \times (r/2)}$. We then define $\max(\mathbf{Z}_{:2q}, \ \mathbf{Z}_{2q:})$ (or either one of $\mathbf{Z}_{:2q}, \ \mathbf{Z}_{2q:}$) as the output of the pooling layer.

Although $\max$ is 1-Lipschitz, its expressive power is limited. For example, consider the 1-Lipschitz function $\|x, y\|_2 = \sqrt{x^2 + y^2}$. It is easy to see that if $x = y$, the error between $\|x, y\|_2$ and $\max(x, y)$ can be arbitrarily large for large values of $x, y$. To address such limitations, we construct expressive pooling layers using the following mathematical property:

**Theorem 2** *Given* $\mathbf{x} \in \mathbb{R}^n$ *and manifold* $\mathcal{M} \subset \mathbb{R}^n$, *the distance function (in* $l_2$ *norm) is 1-Lipschitz:*

$$d_\mathcal{M}(\mathbf{x}) = \min_{\mathbf{x}^* \in \mathcal{M}} \|\mathbf{x}^* - \mathbf{x}\|_2 \implies |d_\mathcal{M}(\mathbf{x}) - d_\mathcal{M}(\mathbf{y})| \leq \|\mathbf{x} - \mathbf{y}\|_2 \qquad (8)$$

The above theorem provides a powerful method for constructing a *learnable pooling layer* by selecting a *learnable manifold* $\mathcal{M}_\Theta \subset \mathbb{R}^r$ ($\Theta$ denotes the set of learnable parameters). To apply the pooling

| Output Size | Layer |
|---|---|
| $32 \times 32 \times 32$ | Conv + MaxMin |
| $64 \times 16 \times 16$ | LipBlock-n/5 |
| $128 \times 8 \times 8$ | LipBlock-n/5 |
| $256 \times 4 \times 4$ | LipBlock-n/5 |
| $512 \times 2 \times 2$ | LipBlock-n/5 |
| $1024 \times 1 \times 1$ | LipBlock-n/5 |
| # of classes | Linear/MLP-1 |

(a) LipConvnet-n Architecture

| Output Size | Layer | Repeats |
|---|---|---|
| $q \times r \times r$ | Input | – |
| $q \times r \times r$ | Conv + MaxMin | $n/5 - 1$ |
| $4q \times (r/2) \times (r/2)$ | Rearrange | 1 |
| $4q \times (r/2) \times (r/2)$ | Conv | 1 |
| $2q \times (r/2) \times (r/2)$ | Pooling | 1 |

(b) LipBlock-n/5

Table 2: LipConvnet-n and LipBlock-n/5 architectures

operation to an input $\mathbf{x} \in \mathbb{R}^n$, we simply output its $l_2$ distance to the manifold $\mathcal{M}_\Theta$ denoted by $d_{\mathcal{M}_\Theta}(\mathbf{x})$. Since $d_{\mathcal{M}_\Theta}(\mathbf{x}) \in \mathbb{R}$ while $\mathbf{x} \in \mathbb{R}^n$, this operation reduces the input size by a factor of $n$.

As an example, let $\mathcal{M}_{\mathbf{u}} = \{\mathbf{u}\}$. Here $\mathbf{u} \in \mathbb{R}^n$ is the learnable parameter and the distance function $d_{\mathcal{M}_{\mathbf{u}}}(\mathbf{x}) = \|\mathbf{x} - \mathbf{u}\|_2$. Even in this very simple case, for $\mathbf{u} = \mathbf{0}$, $d_{\mathcal{M}_{\mathbf{u}}}(\mathbf{x}) = \|\mathbf{x}\|_2$ and for $n = 2$, this function can learn to represent the function $\|x, y\|_2$ discussed earlier exactly. We also prove that a *signed* $l_2$ distance function can be defined when $\mathcal{M}$ satisfies certain properties:

**Corollary 1** *If $\mathcal{M}$ is a connected manifold that divides $\mathbb{R}^n$ into two nonempty connected sets $\mathcal{A}$ and $\mathcal{B}$ such that $\mathcal{A} \cap \mathcal{B} = \phi$ and every path from $\mathbf{a} \in \mathcal{A}$ and $\mathbf{b} \in \mathcal{B}$ intersects a point on $\mathcal{M}$, then there exists a signed $l_2$ distance function with different signs in $\mathcal{A}$ and $\mathcal{B}$.*

Proofs of Theorems 2 and Corollary 1 are given in Appendix A.3 and A.4 respectively.

In practice, computing $d_{\mathcal{M}}$ can be difficult for high-dimensional $\mathcal{M}$. To tackle this issue, we use 2D pooling layers where $\mathcal{M}$ is defined to be a piecewise linear curve in 2D. The $l_2$ distance can then be computed by finding the minimum distance to all the line segments and connecting points. We emphasize that even in this relatively simple case, $\mathcal{M}$ can have large number of parameters and $d_{\mathcal{M}}$ can still be efficient to compute because these individual distances can be computed in parallel. We use $\mathcal{M}_\theta$ defined below (illustrated in Figure 1b, lines $L_1$, $L_2$ correspond to $\phi = +\theta, -\theta$):

$$\mathcal{M}_\theta = \{R(\cos\phi, \ \sin\phi): \quad R \geq 0, \ \phi \in \{+\theta, -\theta\}\}$$

In each colored region (Figure 1b), $d_{\mathcal{M}_\theta}$ can be computed using the following:

$$d_{\mathcal{M}_\theta}\left(R(\cos\alpha, \ \sin\alpha)\right) = \begin{cases} R\sin(\alpha - \theta), & 0 \leq \alpha \leq \theta + \pi/2 \\ -R\sin(\alpha + \theta), & 0 < -\alpha \leq \theta + \pi/2 \\ R, & \text{otherwise} \end{cases} \tag{9}$$

To apply projection pooling on $\mathbf{Z}_{:2q}$, $\mathbf{Z}_{2q:}$ discussed previously, we output $d_{\mathcal{M}_\theta}\left(\mathbf{Z}_{:2q}, \mathbf{Z}_{2q:}\right)$.

# 7 Experiments

We perform experiments under the setting of provably robust image classification on CIFAR-10 and CIFAR-100 datasets. We use the LipConvnet-5, 10, 15, ..., 40 architectures for comparison. We use SOC as the orthogonal convolution and MaxMin as the activation in all architectures. All experiments were performed using 1 NVIDIA GeForce RTX 2080 Ti GPU. All networks were trained for 200 epochs with initial learning rate of 0.1, dropped by a factor of 0.1 after 100 and 150 epochs. For adversarial training with curvature regularization, we use $\rho = 36/255$ (0.1411), $\gamma = 0.5$ for CIFAR-10 and $\rho = 0.2$, $\gamma = 0.75$ for CIFAR-100. We find that certifying robustness using CRC-Lip is computationally expensive for CIFAR-100 due to large number of classes. To address this issue, we only use classes with top-10 logits (instead of all 100 classes) for CIFAR-100 (Table 4). Since CRC-Lip requires us to solve a convex optimization, we consider a certificate to be valid if the input is correctly classified and gradient at the optimal solution is $\leq 10^{-6}$ (0 otherwise). All results are

Table 3: Provable robustness results on CIFAR-10 (LipConvnet-5, 10 results in Appendix Table 5)

| LipConv net- | Methods | Standard Accuracy | Provable Robust Accuracy | | | Increase | |
| --- | --- | --- | --- | --- | --- | --- | --- |
| | | | 36/255 | 72/255 | 108/255 | (standard) | (36/255) |
| 15 | Baseline | 77.78% | 62.75% | 46.34% | 31.38% | _ | _ |
| | + Fast | 77.75% | 62.52% | 46.23% | 31.19% | -0.03% | -0.23% |
| | + CRC | **79.44%** | **66.99%** | **52.56%** | **38.30%** | **+1.66%** | **+4.24%** |
| 20 | Baseline | 77.50% | 63.31% | 46.42% | 31.53% | _ | _ |
| | + Fast | 77.13% | 62.05% | 45.86% | 31.13% | -0.37% | -1.26% |
| | + CRC | **79.13%** | **66.45%** | **52.45%** | **38.12%** | **+1.63%** | **+3.14%** |
| 25 | Baseline | 77.18% | 62.46% | 45.78% | 31.16% | _ | _ |
| | + Fast | 76.94% | 61.91% | 45.59% | 30.69% | -0.24% | -0.55% |
| | + CRC | **79.19%** | **66.28%** | **51.74%** | **37.99%** | **+2.01%** | **+3.82%** |
| 30 | Baseline | 74.43% | 59.65% | 43.76% | 29.16% | _ | _ |
| | + Fast | 74.69% | 58.84% | 43.33% | 28.93% | +0.26% | -0.81% |
| | + CRC | **78.64%** | **66.05%** | **51.31%** | **37.30%** | **+4.21%** | **+6.40%** |
| 35 | Baseline | 72.73% | 57.18% | 42.08% | 28.09% | _ | _ |
| | + Fast | 72.91% | 57.58% | 41.52% | 27.37% | +0.18% | +0.40% |
| | + CRC | **78.57%** | **65.94%** | **52.04%** | **37.63%** | **+5.84%** | **+8.76%** |
| 40 | Baseline | 71.33% | 55.74% | 39.32% | 26.06% | _ | _ |
| | + Fast | 71.60% | 56.15% | 39.82% | 25.63% | +0.27% | +0.41% |
| | + CRC | **78.41%** | **65.51%** | **51.32%** | **37.30%** | **+7.08%** | **+9.77%** |

reported using the complete test sets of CIFAR-10 and CIFAR-100. We compare the provable robust accuracy using 3 different $l_2$ perturbation radii: $36/255, 72/255, 108/255$.

**Table details:** For the baseline ("Baseline" in Tables 3, 4), we use the standard max pooling with the certificate based on LLN [Singla et al., 2022] due to their superior performance over prior works. In Tables 3 and 4, for each architecture, "**+ Fast**" adds faster gradient computation, "**+ CRC**" replaces max pooling with projection pooling (equation (9)) and replaces the last linear layer with a 1-hidden layer MLP (CRC-Lip certificate) while also using faster gradients. For each architecture, the columns "Increase (Standard)" and "Increase (36/255)" denote the increase in standard and provable robust accuracy relative to "Baseline" standard and provable robust accuracy (36/255). Results where Projection pooling and CRC-Lip are added separately are given in Appendix Tables 5 and 6.

**LipConvnet Architecture:** We use a 1-Lipschitz CNN architecture called LipConvnet-n where n is a multiple of 5 and $n + 1$ is the total number of convolution layers. It consists of an initial SOC layer that expands the number of channels from 3 to 32. This is followed by 5 blocks that reduce spatial dimensions (height and width) by half while doubling the number of channels. The architecture is summarized in Tables 2a and 2b. The last layer outputs the class logits and is either a linear layer in which case we use LLN to certify robustness or a single-hidden layer MLP where we use CRC-Lip.

**Correcting the certificates**: Since SOC is an approximation, the Lipschitz constant of each SOC layer can be slightly more than 1 and if we use a large number of SOC layers (e.g. 41 in LipConvnet-40), the Lipschitz constant of the full network ($\mathrm{Lip}(f)$) can be significantly larger than 1. To mitigate this issue, we take the following steps: (a) We use a large number of terms ($k = 15$) during test time to approximate the exponential which results in a small approximation error (using the error bound in Singla and Feizi [2021]) and (b) We compute $\mathrm{Lip}(f)$ by multiplying the lipschitz constant of all SOC layers (using the power method) and then divide the certificate by $\mathrm{Lip}(f)$.

Table 4: Provable robustness results on CIFAR-100 (LipConvnet-5, 10 results in Appendix Table 6)

| LipConv net- | Methods | Standard Accuracy | Provable Robust Accuracy | | | Increase | |
|---|---|---|---|---|---|---|---|
| | | | 36/255 | 72/255 | 108/255 | (standard) | (36/255) |
| 15 | Baseline | 48.06% | 34.52% | 23.08% | 14.70% | _ | _ |
| | **+ Fast** | 47.97% | 33.84% | 22.66% | 14.26% | -0.09% | -0.68% |
| | **+ CRC** | **50.79%** | **37.50%** | **26.16%** | **17.27%** | **+2.73%** | **+2.98%** |
| 20 | Baseline | 47.37% | 33.99% | 23.40% | 14.69% | _ | _ |
| | **+ Fast** | 46.41% | 33.07% | 22.06% | 14.00% | -0.96% | -0.92% |
| | **+ CRC** | **51.84%** | **38.54%** | **27.32%** | **18.53%** | **+4.47%** | **+4.55%** |
| 25 | Baseline | 45.77% | 32.08% | 21.36% | 13.64% | _ | _ |
| | **+ Fast** | 45.28% | 31.67% | 20.69% | 13.26% | -0.49% | -0.41% |
| | **+ CRC** | **51.59%** | **39.27%** | **27.94%** | **19.06%** | **+5.82%** | **+7.19%** |
| 30 | Baseline | 46.39% | 33.08% | 22.02% | 13.77% | _ | _ |
| | **+ Fast** | 45.86% | 32.54% | 21.18% | 12.77% | -0.53% | -0.54% |
| | **+ CRC** | **50.97%** | **38.77%** | **27.73%** | **19.28%** | **+4.58%** | **+5.69%** |
| 35 | Baseline | 43.42% | 30.36% | 19.71% | 12.66% | _ | _ |
| | **+ Fast** | 42.78% | 29.88% | 19.73% | 12.52% | -0.64% | -0.48% |
| | **+ CRC** | **51.42%** | **39.01%** | **28.94%** | **20.29%** | **+8.00%** | **+8.65%** |
| 40 | Baseline | 41.72% | 28.53% | 18.37% | 11.49% | _ | _ |
| | **+ Fast** | 42.07% | 28.51% | 18.86% | 11.89% | +0.35% | -0.02% |
| | **+ CRC** | **50.11%** | **38.69%** | **28.45%** | **20.05%** | **+8.39%** | **+10.16%** |

## 7.1 Results using Faster gradient computation

We show the reduction in training time per epoch (in seconds) on CIFAR-10 in Table 1 and CIFAR-100 in Appendix Table 8. In both Tables, we observe that for deeper networks ($\geq 25$ layers), the reduction in time per epoch is $\approx 30\%$. The corresponding standard and provable robust accuracy numbers are given in Tables 3 (CIFAR-10) and 4 (CIFAR-100) in the row "**+ Fast**". From the columns "Increase (Standard)" and "Increase (36/255)", we observe that the performance is similar to the baseline across all network architectures. For deeper networks: LipConvnet-35, 40, we observe an increase in performance for CIFAR-10 and small decrease ($< 0.64\%$) for CIFAR-100.

## 7.2 Results using projection pooling and CRC-Lip

We observe that using CRC and projection pooling (row "**+ CRC**") leads to significant improvements in performance across all LipConvnet architectures. On CIFAR-10, we observe significant improvements in both the standard ($\geq 1.63\%$, column "Increase (standard)") and provable robust accuracy ($\geq 3.14\%$, column "Increase (36/255)") across all architectures. On CIFAR-100, we also observe significant improvements in the standard ($\geq 2.49\%$) and provable robust accuracy ($\geq 2.27\%$). For deeper networks (LipConvnet-35, 40), on CIFAR-10, we observe even more significant gains in the standard ($\geq 5.84\%$) and provable robust accuracy ($\geq 8.76\%$). Similarly on CIFAR-100, we observe gains of $\geq 8.00\%$ (standard) and $\geq 8.65\%$ (provable robust).

Our results establish a new state-of-the-art for both the standard and provable robust accuracy across all attack radii. In Table 3 (CIFAR-10), the best "Baseline" standard and provable robust accuracy values (at 36/255, 72/255, 108/255) are 77.78% and 63.31%, 46.42%, 31.53% respectively. The best "**+ CRC**" values are 79.57% and 67.13%, 53.17%, 38.60%. This results in improvements of $+1.79\%$ and $+3.82\%$, $+6.75\%$, $+7.07\%$ respectively. Similarly, in Table 4 (CIFAR-100), the best "Baseline" standard and provable robust values are 48.06% and 34.52%, 23.40%, 14.70%. The best "**+ CRC**" values are 51.84% ($+3.78\%$) and 39.27% ($+4.75\%$), 27.94% ($+4.54\%$), 20.29% ($+5.59\%$).

# 8   Acknowledgements

This project was supported in part by NSF CAREER AWARD 1942230, a grant from NIST 60NANB20D134, HR001119S0026 (GARD), ONR YIP award N00014-22-1-2271, Army Grant No. W911NF2120076 and the NSF award CCF2212458.

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
