# OpenReview forum: "Improved techniques for deterministic l2 robustness"
_NeurIPS.cc/2022/Conference — NeurIPS 2022 Accept_

### Official Review · Reviewer_n1ze · 2022-07-10

**Rating:** 5
**Confidence:** 2
**Soundness:** 2 fair
**Presentation:** 2 fair
**Contribution:** 2 fair

**Summary:**

This paper proposes an 1-Lipschitz CNN with improved standard and provable robust accuracies on CIFAR-10 and CIFAR-100 with 1) faster gradient copmutation of SOC layers; 2) 1-hidden layer MLP instaed of last linear layer; 3) 1-Lipschitz pooling layer called projection pooling.

**Questions:**

* This paper ends abruptly without a conclusion section. Is this intended?
* CRC-Lip looks like an effective way to boost performance on 1-Lipschitz CNN, can it scale to larger problems ?
* Comparison only includes LipConvnet, are there any other relavent works not mentioned?

**Limitations:**

Assumptions and Limitations are mentioned in the paper, would be hepful to have a dedicated summarization.

**Strengths And Weaknesses:**

Strengths:
* Originality: The proposed methods for improving 1-Lipschitz CNN on standard and provable robust accuracies is original.
* Quality: The introduced methods for faster gradient computation of SOC layers using approximation, as well as using projection pooling with 1-hidden layer MLP instead of last linear layer is well motivated and justified.
* Clarity: The paper is clearly written with necessary details.

Weakness:
* Significance: The authors mentioned by themsleves that the proposed CRC-Lip for certifying robustness is computationally expensive even for CIFAR-100, this may limit its applicability to broader problems (e.g. ImageNet and etc.)

---

> ### Author Response · Authors · 2022-08-01
> **Author response**
>
> Thank you for your insightful comments.
>
> **Scale to larger problems**: Providing high robustness guarantees for problems with large number of classes is in general a difficult problem and remains an open direction of future research. In this paper, we focus on improving the certificates among classes with top-10 logit values.
>
> **Comparisons other than LipConvnets**: We compare against LipConvnet architectures because they are known to achieve state-of-the-art results for 1-Lipschitz networks.

---

### Official Review · Reviewer_idv1 · 2022-07-11

**Rating:** 5
**Confidence:** 3
**Soundness:** 2 fair
**Presentation:** 3 good
**Contribution:** 2 fair

**Summary:**

This paper presents techniques for accelerating and improving the training of 1-Lipschitz CNNs for certified $\ell_2$ robustness. Specifically, the paper proposes an approximate gradient computation procedure for skew orthogonal convolutions (SOC) which reduces the training time of SOC-based CNNs. The paper also presents a trainable 1-Lipschitz projection-based pooling layer and proposes to replace the last linear layer of 1-Lipschitz CNNs with a certifiable 1-hidden layer MLP for improved robustness. The combination of these techniques improves the state-of-the-art provable $\ell_2$ robustness on CIFAR-10 and CIFAR-100.

**Questions:**

1) How exactly is the certified robust accuracy computed? Is it just the fraction of test inputs whose certificate is smaller than the perturbation strength? It would be better to include this in the main manuscript.

2) Continuing on the pooling layer, it seems to me from Tables 6 and 7 that adding the proposed pooling layers decreases performance. Why is that? and if it is the case then why is it needed? Does it work well in conjunction with the MLP layer? (i.e., is +MLP + pool better than +MLP with standard pool?)

3) Is there a reason why the HouseHolder activation not being used in this work? The authors are aware that it provides improvements in robustness. Does it not stack well with the proposed methods?

4) How long does to take to certify using the proposed methods? and how does it compare to the baselines used in the paper?


**Limitations:**

Paper does not properly address the limitations, e.g. what is the overhead of certification?

**Strengths And Weaknesses:**

Strengths:
- The paper is well motivated and well written. The paper is clearly organized, and the proposed methods are explained pretty well. The theoretical results are clearly stated and all the proofs are provided in the Appendix
- The experimental results seem to back the author's claims, in terms of demonstrating the improvement in certified robustness, however I have some questions regarding the efficacy of the pooling layer (see below)

Weaknesses:
- Going deeper with LipConvnet (as n increases) decreases the natural and robust accuracies consistently, which is counter-intuitive. I understand this is not due to the paper's proposed method, but it makes me wonder why go deeper with these networks in the first place?
- The claim 1-Lipschitz is not technically correct, due to the exponential approximation performed in SOC, is this why going deeper hurts performance?
- The method is restricted to $\ell_2$ norm robustness
- The pooling layer seems to be counter-productive, as it decreases the robustness (see Tables 5 & 6 in the Appendix) in a fairly consistent manner.
- This is a bit unusual, but I have a found a lot of text similarities between this paper and that of the cited work [1]. A decent portion of the text seems to be copied or poorly paraphrased (e.g. Intro, parts of background). I understand the paper seems to be inspired by [1], as the titles are also similar and the methodology (improved last linear layer, improved non-linearity) are aligned. But, in my opinion, this does not justify such 'lazy' writing.


[1] Singla, S., Singla, S., & Feizi, S. (2021, September). Improved deterministic l2 robustness on CIFAR-10 and CIFAR-100. In International Conference on Learning Representations.

---

> ### Author Response · Authors · 2022-08-01
> **Author response**
>
> Going deeper: Constructing high performance lipschitz networks is an active area of research. Since deep networks in general achieve better performance than shallow networks for standard networks, it makes sense to include results for deep lipschitz networks as useful baselines for future research.
>
> **1-Lipschitz claim**: In Line-281 (Page 8), we explain that we divide the certificates with lipschitz constant of the network. Here, the lipschitz constant is computed by multiplying the lipschitz constant of all SOC layers. Thus, all results in the paper are correcting for the lipschitz constant due to the approximation in SOC.
>
> **Restricted to l2 robustness**: Several previous works have shown that a naive conversion of l2 certificates to l_\infty by dividing by \sqrt{d} factor (d is the input dimension) leads to improved state-of-the-art results (such as Page 7, last paragraph in [1]). Due to these reasons, most provable robustness papers focus on l2 robust models.
>
> **Pooling layer**: The goal of the paper is to introduce a rich large class of pooling layers with learnable parameters. The results shown in the paper are using one specific instance of this class and we provide baseline results for the same. A more thorough exploration of pooling layers which lead to the best results will be conducted in a future work.
>
> **Certified robust accuracy**: The certified robust accuracy is the fraction of test inputs whose certificate is larger than the perturbation strength.
>
> **Householder activations**: We did not see a significant improvement in performance due to householder activations and decided not to include it for this reason.
>
> **Time to certify**: The difference between certificates (on CIFAR-10 test set) using LLN and CRC-Lip for LipConvnet-5 networks is as follows:
> 1. Using LLN: 100 seconds
> 2. Using CRC-Lip: 194 seconds
>
> For LipConvnet-40 networks:
> 1. Using LLN: 111 seconds
> 2. Using CRC-Lip: 219 seconds
>
> So indeed, CRC-Lip increases the time to certify but it also improves the certified robust accuracy significantly. However, the time to certify does not increase significantly with the depth of the network. Nevertheless, reducing the time to certify while achieving high certificates is an open problem for future research.
>
> We have made changes in the writing as you suggested.
>
> [1] Provably Robust Deep Learning via Adversarially Trained Smoothed Classifiers. https://arxiv.org/pdf/1906.04584.pdf

---

### Official Review · Reviewer_yBdY · 2022-07-12

**Rating:** 7
**Confidence:** 3
**Soundness:** 3 good
**Presentation:** 4 excellent
**Contribution:** 3 good

**Summary:**

The work proposes 3 methods that could be used to improve the training of the 1-Lipschitz networks from different aspects: (1) an approximation approach for the calculation of gradients in SOC layers which makes the method more time-efficient; (2) changing the final linear layer of a network to an MLP and specifically apply adversarial training on the MLP yields superior performances; (3) a more expressive pooling method compared with max-pooling. The work then empirically demonstrates that using (1) results in time-efficient models with no significant sacrifice in performance and using (2)(3) together boost both natural accuracy and robustness of trained models.

**Questions:**

In the experiment section, it is not entirely clear how the individual components of the proposed methods contribute to the final results. Currently, the message is (1) the fast computation of gradients mostly makes the performances drop; (2) using all 3 proposed methods increases the performances, and (3) from Table 5 and 6 in the Appendix, it seems like using the projection pooling alone also sometimes sacrifices the performances. So does this mean that all the gain is from the usage of MLP? A more detailed ablation study could be done here to better investigate the effect of each individual component.

**Strengths And Weaknesses:**

Strengths:
The overall presentation is clear and easy to follow. The relationships with prior arts are detailly discussed. Also, the empirical results show that the proposed methods are indeed helpful.

Weaknesses:
The reviewer did not find any major flaws in the paper.

---

> ### Author Response · Authors · 2022-08-01
> **Author response**
>
> We have performed more detailed ablation experiments in Appendix Tables 5 and 6. We observe that using fast gradients makes the provable robust accuracy drop by a small amount (~1%) while the standard accuracy remains similar.
>
> Indeed, we observe in our experiments that most performance gains are due to MLP. In this paper, our goal is to introduce a rich large class of pooling layers with learnable parameters (which we call projection pooling). The results shown in the paper are using one specific instance of this class and we provide baseline results for the same. A more thorough exploration of pooling layers which lead to the best results will be conducted in a future work.

---

### Official Review · Reviewer_Rd49 · 2022-07-15

**Rating:** 5
**Confidence:** 3
**Soundness:** 3 good
**Presentation:** 2 fair
**Contribution:** 3 good

**Summary:**

This paper improves and aggregates techniques for 1-Lipschitz CNNs with respect to the $l_2$ norm. First, the authors approximate the gradient for Skew Orthogonal Convolutions (SOC) so the throughput is increased. Second, the paper cooperates Curvature-based Robustness Certificate (CRC) into CNN to replace the last linear layer of the model. Third, it introduces a novel 1-Lipschitz pooling layer which is learnable. Combining these three techniques achieves better standard/robust accuracy on CIFAR-10/CIFAR-100 with a shorter training time.

**Questions:**

- Can you also report the state-of-the-art by other methods, e.g., randomized smoothing?
- The best 36/255 provable robust accuracy on CIFAR-10 is achieved by LipConvnet-5, which is very different from previous results where the best is LipConvnet-20. Do you have any insights into the change?
- In which part of the network do you apply the ``identity map'' in Line 217? Or do you mean that the gradient of the objective is computed as the gradient of $\ell(h_{\Omega}(g_{\Phi}(x) + \epsilon, l)) + \cdots$ where $\epsilon$ is the adversarial perturbation?
- Does "+CRC" rows in tables include faster gradient computation? If not, what is the performance change for "+CRC +Fast"?

Minor:
- Line 49, there is no LipConvnet-$n$ in Table 1.
- Eq (7), there is a missing bracket.
- Line 212, why is $h$ a parameter of $g$?

**Limitations:**

I do not see the negative societal impact of this work.

**Strengths And Weaknesses:**

- This paper provides a good summary of previous techniques and some non-trivial improvements to them.
- The majority of the empirical gain comes from applying CRC to the last layer, which is a decent empirical contribution but not completely novel.
- The new pooling layer is pretty novel as far as I know, but the performance gain is not consistent. Especially for CIFAR-100, the performance drops except for LipConvnet-5. Therefore, I am not sure whether it is a good contribution or not.
- Faster gradient computation for SOC is novel and the speedup brought is notable. However, I am not sure if it is useful in practice because the decrease of provable robust accuracy is not negligible.
- Some writing is not clear enough, especially for readers who are not familiar with previous works. See my questions below. I suggest thorough proofreading.
- Figure 1(a) is very similar to Figure 3 of [Singla and Feizi, 2021]. Please reference.

---

> ### Author Response · Authors · 2022-08-01
> **Author response**
>
> Thank you for your suggestions. We have made changes to the paper as you suggested. Below we provide further clarifications to your comments.
>
> **Lack of novelty of CRC**: The novel component of our work compared to that of prior work is that we perform adversarial training on the penultimate layer of the network as opposed to the input (as used in prior work). This step introduces additional challenges because it requires back-propagation through an adversarial training procedure.
>
> **Performance gains due to pooling**: The goal of the paper is to introduce a rich large class of pooling layers with learnable parameters. The results shown in the paper are using one specific instance of this class and we provide baseline results for the same. A more thorough exploration of pooling layers which lead to the best results will be conducted in a future work.
>
> **Faster SOC gradients (performance drop)**: We note that although faster SOC gradients can lead to ~1% drop in provably robust accuracy in some cases. However in several cases, both the standard and provable robust accuracy improves using faster gradients such as LipConvnet-40 CIFAR-10 (Table 5) and CIFAR-100 (Table 6).
>
> **CRC with faster gradients for SOC**:  Yes, +CRC includes faster gradient computation. We have included the explanation in Page 8, line 271.
>
> **Randomized smoothing for CIFAR-10**: To the best of our knowledge, the best known results for randomized smoothing achieve 60% accuracy at l2 radius 0.25 and 43% at 0.5. Note that we use radii (36/255, 72/155, 108/255) which roughly equal (0.1411, 0.2822, 0.4233). However, we strongly emphasize that results for randomized smoothing are not deterministic and we need to sample ~10^{5} noisy samples per data point to provide a certificate with high probability (i.e. >0.999). Since our work provides deterministic certificates, randomized smoothing is not comparable to our work.
>
> **Best 36/255 robust accuracy**: We believe that the key reason LipConvnet-5 achieves higher provable robust accuracy is because we use the heuristic that the gradient of the adversarial training procedure (applied on the penultimate layer) is an identity map. Thus, for deeper networks, the gains are not as significant. We believe that improving the gradient flow through the penultimate layer can improve performance of deeper networks and remains an open direction of research.
>
> **Identity map**: We take the jacobian of adversarial training to be an identity map and show that this works well in practice.
>
> **h parameter of g**: Due to a missing bracket, there was a confusion. Indeed, h is not a parameter of g.
>
> We have made the changes in the writing as you suggested.

---

### Meta-Review · Area_Chair_HgSv · 2022-08-26

**Recommendation:** Accept
**Confidence:** Certain

**Metareview:**

This paper introduces several new techniques to improve the $\ell_2$ adversarial robustness of CNNs, including approximating the gradient for Skew Orthogonal Convolutions (SOC), replacing the final linear layer with a 1-hidden-layer MLP, and introducing a new class of pooling layers. These techniques lead to improved efficiency and robust accuracies on CIFAR-10 and 100.

All reviewers agree that this paper has made interesting and solid contributions, and find the paper well-written. Therefore, I recommend it to be accepted to the conference.

**Award:**

No

---

### Decision · Program_Chairs · 2022-09-14

Accept